# Analysis of the Different Approaches Adopted in the Italian Regions to Care for Patients Affected by COVID-19

**DOI:** 10.3390/ijerph18030848

**Published:** 2021-01-20

**Authors:** Fabrizio Pecoraro, Daniela Luzi, Fabrizio Clemente

**Affiliations:** 1Institute for Research on Population and Social Policies, National Research Council, Via Palestro, 32, 00185 Rome, Italy; d.luzi@irpps.cnr.it; 2Institute of Crystallography, National Research Council, Via Salaria Km 29300, 00016 Monterotondo, Rome, Italy; fabrizio.clemente@ic.cnr.it

**Keywords:** hospital care, home-based care, epidemic outbreak phases, Italian regional health systems, COVID-19

## Abstract

As the Italian health system is regionally based, COVID-19 emergency actions are based on a general lockdown imposed by national authority and then management at local level by 21 regional authorities. Therefore, the pandemic response plan developed by each region led to different approaches. The aim of this paper is to analyze whether differences in patient management may have influenced the local course of the epidemic. The analysis on the 21 Italian regions considers the strategies adopted in terms of hospitalization, treatment in the ICU and at home. Moreover, an in-depth analysis was carried out on: Lombardia, which adopted a hospitalization approach; Veneto, which tended to confine patients at home; and Emilia Romagna, which adopted a mixed hospitalization-home based approach. The majority of regions implemented a home-based approach, while the hospital approach was followed in three regions (Lombardia, Piemonte, and Lazio), mainly limited to the first period of the outbreak. All regions in the later phases tended to reduce hospitalization, preferring to confine patients at home. This comparison, highlighting the different phases of the pandemic, outlined that the adoption of home-based practices contributed to limiting infection rates among patients and health professionals as well as decreasing the number of deaths.

## 1. Introduction

The number of individuals infected with severe acute respiratory syndrome coronavirus 2 (SARS-CoV-2), the virus causing coronavirus disease 2019 (COVID-19), is spreading dramatically across the globe since its outbreak in China [1] being declared a pandemic by the WHO on 11 March 2020. In Europe, the first tested and declared positive case for SARS-CoV-2 was managed by the hospital of Codogno, Italy on 20 February 2020. From these cases, a rapidly increasing number of patients have been identified, initially in Northern Italy and later in the rest of the country and Europe. During the first weeks of the epidemic spread, Italy was the most affected country in the Western world. The spread of this epidemic virus during these two months has not followed a uniform pattern on the Italian territory. Of the cumulative confirmed cases, 75% occurred before 29 April, with inhabitants of the northern regions counting around 45% of the total Italian population, and the Lombardia region more than 30% of the cumulative cases (around 17% of the Italian population). To date the cumulative number of cases, recoveries and deaths worldwide are respectively around 95, 52 and 2 millions. Among them, Italy remains one of the countries with a higher impact of the COVID-19 with around 2.38 millions of people infected, 1.75 millions recovered and more than 82 thousand of deaths. Although the number of cumulative cases is more similarly distributed among regions, in Italy the spread of the virus remains prevalent in the northern regions (more than 60% of all cases) and, in particular, in Lombardy (more than 17% of all cases).

To contain the spread of the virus and facilitate the treatment of more serious patients, Italy developed an overriding national strategy for COVID-19 response that resulted in a national lockdown, which started on the 9/10 March 2020. Within these broad guidelines, each region developed a response plan to manage patients on the basis of strengths and weaknesses of its curative and public health services [2,3], the availabilities of structural and professional components and the amount of cases clarified during the first days of the outbreak. These regional plans were enforced due to the regionally-based Italian health systems which makes local authorities responsible for the organization and delivery of health services. 

Among the plethora of scientific and newspaper articles published in these months on coronavirus, the different organizational approaches adopted by the Italian regions to care patients have been analyzed in a few papers [2,4,5,6,7] focusing the attention on the differences between Lombardia and Veneto regions. As reported by Pisano et al. [2], the organization of health services in Lombardy and Veneto has influenced the approaches taken in the critical early weeks of the epidemic. While Lombardy based the care model on a large hospitalization relying mainly on its network of public and private hospital structures, Veneto implemented a broad community-based strategy that relied on its more robust public health network and local integration of services [2]. Another important analysis conducted by Binkin et al. [6] suggests that the community-based approach implemented in Veneto appears to be associated with substantially reduced rates of cases, hospitalizations, deaths, and infection of healthcare workers compared with the hospital-centered approach adopted in Lombardy. Note that, the results of both studies are based on data collected at the end of March/beginning of April and do not consider the evolution of the epidemic. 

Compared to the above-mentioned literature, this study extends the analysis to the 21 Italian regions considering the hospitalization rate of patients affected by COVID-19 over a larger period of time on the basis of data collected and published by the ISS (Italian National Institute of Health) until the 29 April. Practically, the maximum alert phase was officially closed on the 3 May by the national authorities and Italy entered the second phase of its lockdown the day after. This analysis models data collected within the pandemic phases of initiation, acceleration, and deceleration, as described by the WHO [8] and the US CDC [9]. This distinction helps correlate the different phases with the patient management approach adopted in each region aiming to explore whether and to what extent the organization of health delivery has had an impact on the subsequent phases in terms of containment and monitoring of cases. This may also outline lessons learned prompting appropriate changes in the future. 

## 2. Materials and Methods

### 2.1. Data Sources 

This paper is focused on two main data sources. The first one [10] provides daily updated data on the virus diffusion during the COVID-19 outbreak as reported by the official government website [11]. It exposes official and continuously updated information produced by the Italian Civil Protection Department and was adopted by the Ministry of Health for its periodic report. In particular, the attention is posed on the following variables: Number of diagnosed casesNumber of patients affected by COVID-19 that are: hospitalized in a ward, hospitalized in an ICU, and confined to homeNumber of deathsNumber of recoveries

All variables are collected both as daily cases and as cumulative cases over the period. In this study the analysis is performed considering the period from 24 February to 29 April (i.e., four days before the Italian Government indicated the conclusion of the so-called lockdown phase 1). 

The second source of data is the database of the Italian Statistics Institute (ISTAT) [12], which is the main producer of official statistics in Italy including the census of population. From this database, we collected the resident population of each Italian region to normalize the above-mentioned variables and provide a comparable analysis across regions. 

The entire dataset as well as the relevant analyses performed during the current study are available in the Zenodo repository [13].

### 2.2. Data Modelling in Identification of Epidemic Phases

The cumulative number of COVID-19 cases that occurred in each region have been collected to identify the different phases of the epidemic outbreak. For a finalized epidemic, the curve for the cumulative number of infected cases has been fitted using the logistic growth regression. This model has been adopted in different studies both to represent the COVID-19 outbreak [14,15,16,17] and to analyse patterns of the epidemic occuring in the past [18]. In particular, in this paper we adopted the following expression of a logistic function:C(t) = K/(1 + Ae^(−rt)) (1)
where: C(t) represents the cumulative number of cases at a given time t;K is the maximum number of cumulative cases K (to be) reached by the relevant region at the end of the epidemic;A describes the position of the curve with respect to the x axis;r indicates the slope of the curve in the acceleration and deceleration phases of the epidemic period.
t_p_ = (ln(A))/r(2)

These parameters are used to identify the following curve points of interest and, as a consequence, to capture the different phases of the epidemic period. In particular, the inflection point computed using Formula (2) represents the point when the number of daily cases starts to decrease, and the cumulative cases start to decelerate, and the infection starts to become less aggressive. At this day, the number of cumulative cases reached the K/2 value. Starting from the inflection point (see Formula (2)) and taking the slope of the curve (r) into account, the following phases can be detected (Figure 1): Initiation for t < t_p_ − (2/r), when the number of cases slowly, exponentially increase.Acceleration for t_p_ − 2/r < t < t_p_ when the number of cases rapidly increase over time with an aggressive infection spread.Deceleration for t_p_ < t < t_p_ + 2/r when the number of daily cases start decreasing and the cumulative cases start decelerating and the infection starts becoming less aggressive.Preparation for t > t_p_ + 2/r when few number of daily cases are diagnosed and the curve tend to K.

The fast-growing period of the infection that comprises the acceleration and deceleration phases is 4/r.

In particular this work focused the attention on the first three phases of the epidemic to verify not only the state of each region in terms of number of cases, deaths, and recoveries, but also to capture whether and to what extent the organizational approach adopted in each phase can influence the outcome of the subsequent phases.

## 3. Results

### 3.1. Data Analysis 

Figure 2 reports the cumulative number of cases diagnosed in each region starting from 24 February (day 1) to 28 April (day 65). All trajectories are reported normalizing the number of cases per 100,000 resident population and using a coherent scale to highlight the different impact the virus had throughout the country. 

Starting from these data, the first evident result is that the Italian territory can be divided into three main clusters of regions depending on the number of coronavirus cases: Regions with low prevalence (lower than 33% of the maximum number of cases per 100,000 inhabitants): Calabria, Sicilia, Basilicata, Campania, Sardegna, Molise, Puglia, Lazio, Umbria, Abruzzo, Friuli Venezia Giulia, Toscana;Regions with medium prevalence (between 33% and 66% of the maximum number of cases per 100,000 inhabitants): Veneto, Marche, P.A. Bolzano, Liguria, Emilia-Romagna, Piemonte;Regions with high prevalence (higher than 66% of the maximum number of cases per 100,000 inhabitants): P.A. Trento, Lombardia, Valle d’Aosta.

The different epidemic phases are subsequently assessed adopting the methodology described in the previous paragraph. The start and end day of the single phases as well as the parameters of the logistics curves are reported in [13]. Note that significant results in the fitting procedures were obtained with the coefficients of determination (R squared) higher than 0.99 for all trajectories.

### 3.2. Application of the Model and Patient Management 

Starting from the results of the previous section, we further analyse the organisational approach adopted by each region in terms of hospitalization and home care. Figure 3 shows the number of cumulative hospitalized cases (red curve), the ones isolated at home (blue) and the inpatients in the ICU (green) in each Italian region. The three vertical lines identifies the transition between the different phases of the epidemic as computed in the previous paragraph. In particular, the violet line specifies the transition between the initiation and the acceleration phases, the azure one between the acceleration and the deceleration phases and the orange one between the deceleration and the preparation phases. 

On the basis of the trajectories shown in Figure 3 in each epidemic phase, the following four distinct approaches can be detected: Patients treated predominantly at home even from the initial phase. This pattern is even more evident during the subsequent phases. Regions that belong to this group are: Campania, Friuli, P.A. Bolzano, P.A. Trento, Sardegna, Toscana, Umbria, Val d’Aosta, Veneto. Among these regions, Veneto represents an important case study as the home care approach is adopted already during the first days of contagion.Mixed approach where nearly half of the patients are treated in the hospital and the other half are confined at home. This is shown particularly during the first phase, while subsequently the home care approach progressively increases during the second and, even more, during the third phase. Regions within this group are: Abruzzo, Basilicata, Calabria, Emilia Romagna, Marche, Puglia. Among them, Emilia Romagna represents an interesting case study for two main reasons: the number of cases and the home care pattern is quite similar to the one of the Veneto region during the second and third phases.Similar to the previous one, this group adopts a mixed approach also in the second phase of the infection. Home care began to be prevalent during the third phase. Regions that adopt this pattern are: Liguria, Molise, Sicilia.Patients treated predominantly at hospital in particular during the initial phase. Subsequently, these regions adopt a mixed approach during the second phase and then a home care approach during the third phase. Regions within this group are: Lazio, Lombardia, Piemonte. Among them, Lombardia represents an interesting and widely analyzed case study considering in particular the high ratio between hospital and home care as well as the fact that the mixed approach is adopted only during the second part of the acceleration phase. Similarly, this region tends to hospitalize an important number of patients also during the third phase with a predominant home care model adopted only at the end of the deceleration phase.

The variables captured at the points of phase transition are reported in Table 1 for the selected use case regions, while the results for the entire set of regions are reported in the Appendix A and in the repository along with the raw data [13]. Variables reported in the table are normalized per 100,000 inhabitants. 

Among the variables captured and presented in Table 1, the attention is focused on two main composite variables computed in the three distinct epidemic phases: the ratio between hospitalized patients and those confined at home to specify the approach adopted by each region in term of level of hospitalization vs. home care;the death rate as well as the recoveries to capture the impact of the virus in the specific region.

In the following, a comparison of the approaches adopted in the three regions in the different epidemic phases is reported considering the above-mentioned composite indicators. 

Initiation phase: as outlined previously, during the first part of the epidemic outbreak in Italy, Lombardia, Veneto, and Emilia Romagna managed patients with coronavirus adopting three different approaches. This is evident considering the rate between the ward and the home treated cases: while in Veneto the majority of patients were confined at home (N = 26.29 per 100,000 inhabitants) with a low involvement of hospital facilities (N = 7.46 in the wards and N = 2.43 in the ICU), both Emilia Romagna and Lombardia treated a significant number of individuals in the hospital structures. In particular, while the Emilia Romagna adopted a mixed approach balancing the high number of inpatients with those treated at home, the number of admissions in Lombardia was five times higher than the number of individuals treated at home and, in addition, with a consistent number of patients treated in the ICU (N = 4.65). During this phase, the death rate remained relatively low in all regions with around five deaths per 100,000 inhabitants in Lombardia and Emilia Romagna and only one death in Veneto. On the contrary, a significant number of patients that had already recovered from the virus were detected in Lombardia (around nine individuals per 100,000 inhabitants) with respect to Emilia Romagna (N = 1.15) and Veneto (N = 2.18). The high level of recoveries in Lombardia can be explained by the fact that home diagnostic tests may have longer response times than those performed in the hospital. 

Acceleration phase: the approaches adopted by the three regions in the initiation phase did not change when the number of cases started to increase exponentially. In fact, although Lombardia had a hospitalization rate lower than one, meaning the proportion of patients confined at home were higher than those treated in the hospital wards, the high death rate (N = 54.61) mainly occurred in the hospital structures. This is evident considering the ratio between the total number of hospital and home days spent by the patients during this phase is still higher than 1.2 in this region. Of course, there was a high number of individuals confined at home with respect to the initiation phase, meaning that also Lombardia started providing home services to patients with low risk symptoms. A positive trend can be detected in the Emilia Romagna region with a hospitalization rate around 0.5. This tendency is also confirmed considering the total number of hospital and home days. However, in this region, a consistent number of deaths can also be detected (N = 27.9), which is even higher than the recovery rate (N = 24.49). The Veneto region continued to provide a home care approach with the majority of the patients treated at home, also having the lowest level of death rate among the three regions. 

Deceleration phase: as previously mentioned in the deceleration phase, all regions had finally adopted a home care approach with a significant reduction of patients treated in hospitals. This is evident not only in Veneto and Emilia Romagna that continued their trend, but also in Lombardia with the hospitalization rate set to 0.13 and the total number of hospital bed days lower than the total number of home days. However, it is important to note that Lombardia is the only region that continued increasing the number of inpatients during this phase (N = 9.23) compared to Veneto (N = −6.54) and Emilia Romagna (N = −6.65).

## 4. Limitations of the Study 

The approach adopted in each region is strongly coupled with the testing strategies put in place to capture the majority of patients affected by the virus. The extensive testing and contact tracing provided over the territory can identify and control a wide number of infected patients, reducing possible transmissions to other individuals [19,20]. 

As a matter of fact, while in Lombardy and Emilia Romagna more than 25% of the swabs carried out were positive, in Veneto this rate dropped to 7% (Appendix A). This clearly indicates that two specific strategies were adopted by these regions: Lombardy and Emilia Romagna provided this test mainly to symptomatic patients, while Veneto tended to test not only those patients with symptoms attributable to COVID-19, but also asymptomatic individuals. This aspect is clearly important considering that the great majority of people infected are asymptomatic, but they represent one of the main sources of contagion in particular among those individuals who work closely and might pass the virus to colleagues, patients, or relatives in case of health personnel. The wider adoption of swabs performed in the Veneto region has certainly helped the virus containment and could have influenced the ratio between the number of days spent in ward and those spent at home. However, we consider it as one of the possible determinants of outcomes which should need further and more comprehensive analyses. 

Another important factor to be considered is the high number of deaths due to COVID-19 that occurred in the nursing homes in Italy [21] as well as in many other countries [22]. It is estimated that, in the first period of the epidemics (from March to May 2020) deaths in nursing homes represent 32.2% of the total number of COVID-19 deaths in Italy [23]. Even though these findings are only partially reliable, they clearly show that they significantly contributed to the total amount of COVID-19 deaths in Italy. However, official data may lack the inclusion of patients that died in nursing homes in its statistics if they had not received a microbiological diagnosis. This makes it necessary to further analyze data on cases and deaths to capture the impact of COVID-19 in nursing homes at a regional level. 

Moreover, in this paper, the comparison between these regions did not consider differences in population density and social factors, as well as the higher initial number of cases in Lombardy and higher numbers of initial foci that may have played a role in the observed differences in outcomes [6]. As said in the discussion, important information when analysing these patterns is the availability of hospital beds and professionals even before COVID emergency, in which the number largely has been increased and not well monitored [24]. Moreover, intra-regional differences might be further analysed to capture the influence of health structural components in the diffusion of the virus [25] and the efficacy of the outbreak treatment [26] including the interpretation of the death rate in light of the demographic context [27]. Finally, the different approaches adopted in other countries may be investigated to capture their effect on the virus transmission and death rate. However, data are heterogeneously collected across countries, making it difficult to compare the past and the actual situation in particular at considering the hospitalization rate at regional level [28]. 

## 5. Conclusions 

In this paper we extended the analysis to all Italian regions capturing the different approaches adopted to treat patients with coronavirus considering the three phases of the epidemic process. The onset day of each phase as well as its duration have been captured through a regression analysis fitting the cumulative number of cases with a logistic model. This made it possible to compare the approach adopted by each region depending on the different epidemic phase on the basis of the different time lines of the course of COVID-19 infection. 

The study is based on the 21 Italian regions considering data analysis unbiased from the different policies for social segregation imposed at national level. It outlines that: the majority of the regions (N = 9; 43%) adopted a home-based approach, while the hospital approach has been implemented only in three regions (14%, Lombardia, Piemonte, and Lazio) and was mainly limited to the first epidemic phases. The remaining regions provided a mixed hospital-home based approach (N = 9; 43%). Moreover, it is evident that, despite different approaches adopted in the first phases of the epidemic, all regions in the later phases tended to reduce hospitalization and preferred to confine patients at home. 

Specific differences can be detected, such as in Lombardia, where this switch between hospital and home care apparently happened only in the second part of the deceleration phase. When analysing these patterns, the availability of hospital beds and professionals which varies across regions is important information to consider. However, even if the number of both hospital beds and health professionals appointed for COVID-19 has been largely increased in this epidemic period [24], data are not available yet and unfortunately were also poorly monitored even before the outbreak.

The majority of the regions, in particular in the central-southern regions, that applied the mixed approach did not saturate the hospital structures, even if the number of inpatients increased over time. For this reason, to provide an in-depth analysis of these approaches, we subsequently focused the attention on the three regions with the highest number of cases: Veneto, Lombardia, and Emilia Romagna. In particular, we performed the analysis considering the following variables: the hospitalization rate in terms of the number of patients treated in the hospital wards and those confined at home, the number of deaths, and the number of recoveries. 

This study confirms the results previously published, highlighting that the hospitalization approach adopted in Lombardia may have influenced the diffusion of the virus in the hospital structures as well as the death rate affecting healthcare workers, patients already hospitalized for other pathologies and healthy individuals [6,18,29]. Whereas region with a lower hospitalization rate, even with medium prevalence, show a lower level of active cases and deaths (see Table 1).

This approach could have been influenced by the sudden outbreak of the virus as well as by the limited knowledge of its treatment that might have led to admitting patients with relatively modest symptoms, in particular in Lombardia during the first phase (i.e., acceleration) of the outbreak [18]. On the contrary, although the outbreak started almost in the same period, the Veneto region tended to confine the patients in their home setting instead of treating them in the hospital. 

The strategies adopted by each region in particular during the acceleration phase of the outbreak could have impacted the infection rate among patients and health professionals as well as on the workload of the hospitals that, in some regions, have led the local governments to invest to increase the number of beds available over the territory. The adoption of efficient and effective strategies is particularly crucial during the first stages of the epidemic period, considering that, in the first weeks, on average the length of stay of a patient affected by COVID-19 was about 19 days. Thus, a substantial number of hospitalizations in the first days not only influences the occupancy rate in the later stages of the epidemic with a critical overuse of the hospital structural and personnel components, but also impacts the organizational workload and on the different capacities of the regional systems in terms of number of hospitals, beds, doctors, nurses and midwifes, and territorial services. 

Moreover, a lower hospitalization rate may have contributed to limiting the contact with health care settings and thus with more fragile patients and health care workers. This confirms that, despite the uncertainty about what is the optimal strategy to tackle the current pandemic, the territorial management model is one of the best responses to face this emergency [26]. Of course, this not only requires strong formal territorial health services (i.e., primary care physicians, territorial nurses) but also necessitates the participation and reliance of informal home healthcare providers (i.e., family) who play an important role in supporting patient’s specific needs [30].

## Figures and Tables

**Figure 1 ijerph-18-00848-f001:**
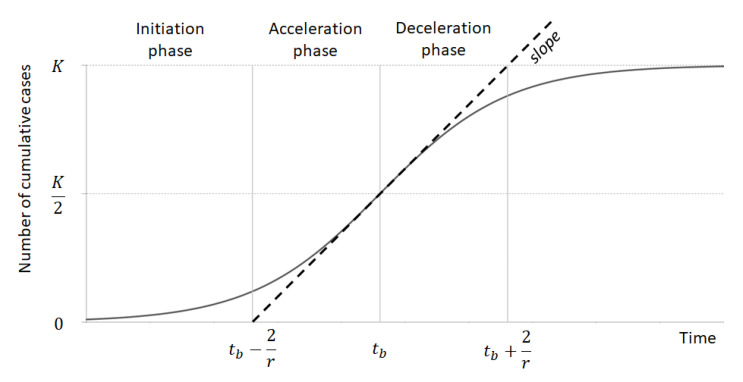
Example of a logistic curve representing the cumulative number of cases assessed over time (modified from [16]).

**Figure 2 ijerph-18-00848-f002:**
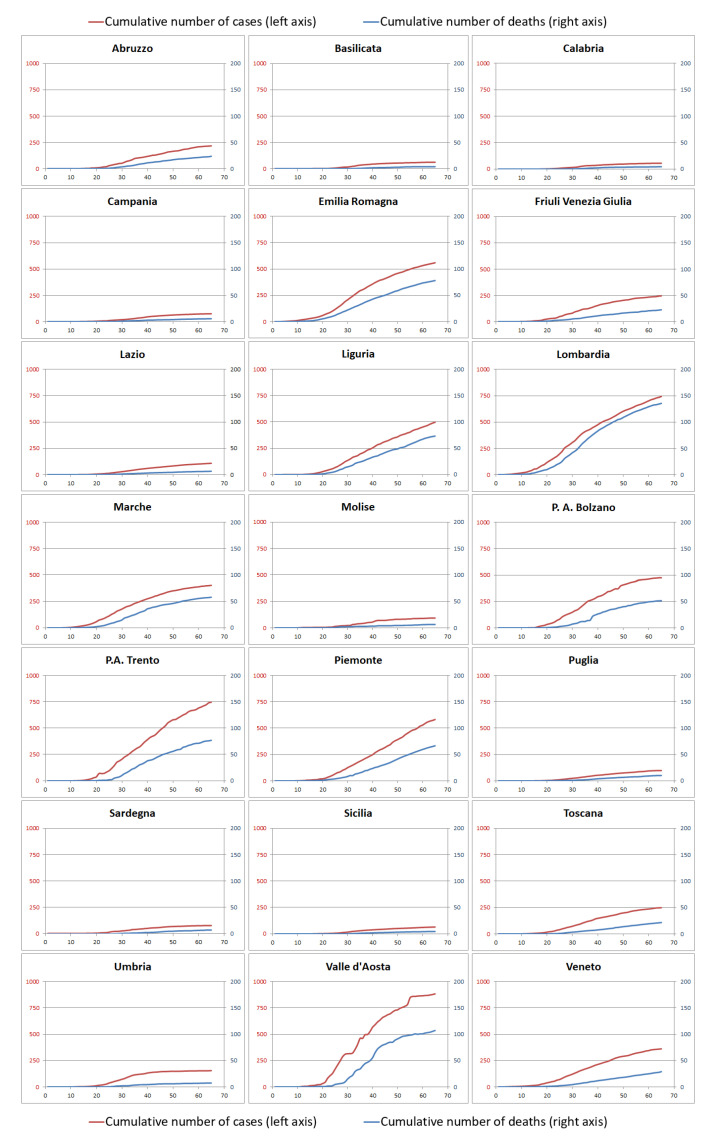
Cumulative number of cases (red lines and left scales) and deaths (blue lines and right scales) over time in each region per 100,000 inhabitants.

**Figure 3 ijerph-18-00848-f003:**
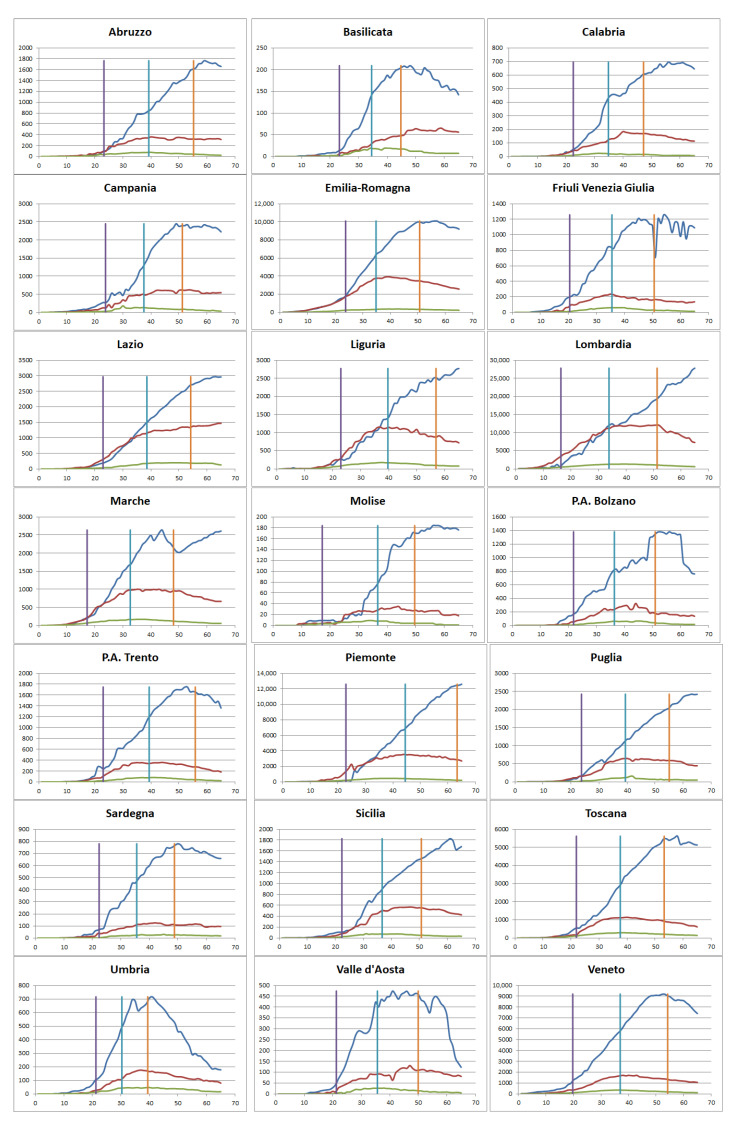
Cumulative number of cases over time hospitalized (red line), confined at home (blue line) and treated in the ICU (green line). Vertical lines identify the transition between the different epidemic phases: the violet line specifies the transition between the initiation and the acceleration phases, the azure one between the acceleration and the deceleration phases and the orange one between the deceleration and the preparation phases.

**Table 1 ijerph-18-00848-t001:** Measures captured in Lombardia, Emilia Romagna, and Veneto during the three phases of the epidemic. Data are normalized per 100,000 inhabitants. In particular: number of patients treated in the hospital (ward), ICU and confined at home along with the relevant total number of days spent by the patients; the ratio between the number of ward and home cases; number of active cases, deaths and recoveries; cumulative number of cases.

Region	Start: End Day	Ward (Days)	ICU (Days)	Home (Days)	Ward/Home (Days)	Active	Deaths	Recoveries	Total Cases
Emilia Romagna	1:19	21.16 (144.18)	2.88 (19.43)	21.14 (142.51)	1.00 (1.01)	45.18	4.52	1.15	50.85
Lombardia	1:16	33.1 (161.01)	4.65 (30.55)	6.4 (63.97)	5.17 (2.52)	44.15	4.67	8.94	57.76
Veneto	1:20	7.46 (52.14)	2.43 (15.33)	26.29 (149.97)	0.28 (0.35)	36.18	1.12	2.18	39.48
Emilia Romagna	20:35	63.52 (875.38)	4.6 (93.65)	123.4 (1256.21)	0.51 (0.70)	191.52	27.9	24.49	243.91
Lombardia	17:34	78.11 (1377.51)	8.5 (176.69)	113.65 (1131.25)	0.69 (1.22)	200.26	54.61	80.43	335.3
Veneto	21:37	26.78 (390.50)	4.83 (93.27)	92.21 (1225.38)	0.29 (0.32)	123.82	8.6	14.7	147.12
Emilia Romagna	36:51	−6.65 (1339.63)	−0.11 (127.66)	79.62 (3098.27)	−0.08 (0.43)	72.86	28.36	70.28	171.5
Lombardia	35:51	9.23 (2014.93)	−1.96 (215.97)	71.06 (2540.64)	0.13 (0.79)	78.32	51.83	88.35	218.51
Veneto	38:54	−6.54 (535.25)	−3.24 (96.22)	66.2 (2839.70)	−0.10 (0.19)	56.42	11.19	59.14	126.75

## Data Availability

The entire dataset as well as the relevant analyses performed during the current study are available in the Zenodo repository, DOI: 10.5281/zenodo.3865424.

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
