# Peer review of "Analysis of the Different Approaches Adopted in the Italian Regions to Care for Patients Affected by COVID-19"

_ijerph, 2021, doi:10.3390/ijerph18030848_

Round 1
Reviewer 1 Report
The submitted article is a brilliant and very interesting analysis of the modalities of the 21 regional health systems in answer to the SARS-COV-2 pandemic in the period of its greatest spread in Italy. The authors have used the official data provided by the National Department of Civil Protection and those of the Italian Institute of Statistics for population data. The proposed logistic model made it possible to clearly identify the three phases of the infection trend in the population, differentiating the specific characteristics for the different regions. The analysis thus identified almost 3 different regional behavioral patterns in the prevalent management of infected patients (home care, hospitalization, and mixed). The authors analyze and discuss the 3 models in more detail by examining for each approach of care the most representative regions with the largest population. The conclusions emerging from the analysis are sequential to the approaches taken by the different regional systems, in particular highlighting how the hospital approach appears to be the least favorable for containing the spread of the virus.
Nevertheless, the authors should discuss and attempt to clarify some data in light of some additional specifications.
- As the authors also report, the recognition of cases in different regions is related to a different approach to performing diagnostic swabs. Some regions only swabbed patients with the most severe symptoms coming to hospital observation, whereas in other regions diagnostic swabs were also performed on asymptomatic or paucisymptomatic populations. could there be a possibility of an underestimation of the number of cases making the obtained results less robust?
- Another point to be discussed, at least about the spread of cases and deaths, was the management of elderly residents in nursing homes, exposed to infection due to the simultaneous presence of Covid positive patients, in some regions and not in other ones.
- Could the number of deaths be affected by the lack of a laboratory diagnosis of Covid-19? Many deaths in places and periods of increased virus spread, due to lack of swab diagnosis, maybe underestimated, while deaths in terminal patients may have been attributed to Covid-19.
- Have the authors evaluated whether the territorial medicine in the different regions has had a different impact on the number of recognized cases in the function of the organization and the availability of nursing and medical staff and adequate diagnostic tools and personal protective equipment?
- Based on the available data, what hypotheses can the authors formulate which interpret most fully the difference in the number and severity of events between regions with low "incidence" (i.e. Basilicata, Campania, Calabria) and those with the highest (i.e. Lombardy, Piedmont, Trento): relationships between public and private health systems, territorial health organization, geographical location, social structure, population density, other.
Author Response
Dear Reviewer,
We thank you for revising our paper submitted to the International Journal of Environmental Research and Public Health. We are grateful for your suggestions and comments.
We have answered the comments raised in a point-by-point format addressing any concerns. Comments are reported in red in the following. The manuscript has considerably improved by these modifications and we thank you for your attention. We have highlighted the changes in the main text using track changes in Word.
Yours Sincerely,
Fabrizio Pecoraro
The submitted article is a brilliant and very interesting analysis of the modalities of the 21 regional health systems in answer to theSARS-COV-2 pandemic in the period of its greatest spread in Italy. The authors have used the official data provided by the National Department of Civil Protection and those of the Italian Institute of Statistics for population data. The proposed logistic model made it possible to clearly identify the three phases of the infection trend in the population, differentiating the specific characteristics for the different regions. The analysis thus identified almost 3 different regional behavioural patterns in the prevalent management of infected patients (home care, hospitalization, and mixed). The authors analyse and discuss the 3 models in more detail by examining for each approach of care the most representative regions with the largest population. The conclusions emerging from the analysis are sequential to the approaches taken by the different regional systems, in particular highlighting how the hospital approach appears to be the least favourable for containing the spread of the virus. Nevertheless, the authors should discuss and attempt to clarify some data in light of some additional specifications.
- As the authors also report, the recognition of cases in different regions is related to a different approach to performing diagnostic swabs. Some regions only swabbed patients with the most severe symptoms coming to hospital observation, whereas in other regions diagnostic swabs were also performed on asymptomatic or paucisymptomatic populations. could there be a possibility of an underestimation of the number of cases making the obtained results less robust?
As reported in the limitation of the study paragraph, the testing strategies may be strongly coupled with the approach adopted in each region (home care vs. hospital care). However, this only partially influenced the results reported in this paper. Considering for instance Veneto and Lombardia, the number of swabs (per 100.000 population) performed at the end of the acceleration phase were: Lombardia: 807,99 and Veneto: 1615,43. This clearly highlights that Veneto performed double of swabs carried out in Lombardia. Even if this can have influenced the ratio between the number of days spent in ward and those spent at home, the Lombardia region has a value around four times higher than the Veneto region (see Table in Appendix. Lombardia: 1,22 and Veneto: 0,32) highlighting that this aspect is only one of the possible determinants that had influenced the results reported in our study. It is interesting to note that (see Table 1 in the manuscript) this difference between the two regions is mainly due to the high number of patients hospitalized in Lombardia (1377,51 days per 100.000 population) compared to Veneto (390,50 days per 100.000 population), while the total number of home days were similar (1131,25 Lombardia and 1225,38 Veneto). The text of the manuscript has been updated in the discussion and conclusion paragraph taking into account this consideration.
- Another point to be discussed, at least about the spread of cases and deaths, was the management of elderly residents in nursing homes, exposed to infection due to the simultaneous presence of Covid positive patients, in some regions and not in other ones.
Prevention of COVID 19 infection in Italian Care Homes was poor, especially in the first phase of the virus. This had clearly influenced the spread of the virus as well as the mortality in the nursing homes of both professionals and patients. Italian nursing homes were probably the first residential institutions in the western world that have been hit by the pandemic (Arlotti and Ranci, Report: The Impact of COVID-19 on nursing homes in Italy, 2020. Available from: http://www.lps.polimi.it/wp-content/uploads/2020/06/Nursinghomes_inage.pdf). The same situation happened in many other countries (Comas-Herrera et al., Mortality associated with COVID-19 outbreaks in care homes: early international evidence, 2020, https://ltccovid.org/wp-content/uploads/2020/06/Mortality-associated-with-COVID-21-May-1.pdf) in later times, such as in Belgium. According to the results of a National Survey carried out by the Istituto Superiore di Sanità (Lombardo et al., The Italian national survey on Coronavirus disease 2019 epidemic spread in nursing homes, 2020. Available from: https://onlinelibrary.wiley.com/doi/pdf/10.1002/gps.5487?casa_token=bztuvrC8S7MAAAAA%3A7i5jSWR1gD9M0mFASNvkb0hQosLuAr9KZUhZjPH1QLcDB7o_6EPlLOV8wsXD6-3Feg44bhUa3iJf4w), the mortality rate due to Covid-19 was calculated to be 3.3% at national level, but it rose to 6.7% in Lombardia. However, it is difficult to implement an in-depth analysis given that in Italy official data on COVID-19 does not include casualties in nursing homes in its statistics. If patients died in these facilities without having received a microbiological diagnosis, they might not be recorded by official records of COVID-19 victims, thus determining an important gap of undetected deaths. A previous study (Pesaresi, Il Covid-19 nelle strutture residenziali per anziani, “I luoghi dell cura”, 2020. https://www.luoghicura.it/dati-e-tendenze/2020/05/il-covid-19-nelle-strutture-residenziali-per-anziani/) estimated that, in the period March-May 2020 deaths in nursing homes represent 32,2% of the total number of covid-19 deaths in the country (Arlotti and Ranci, Report: The Impact of COVID-19 on nursing homes in Italy, 2020. Available from: http://www.lps.polimi.it/wp-content/uploads/2020/06/Nursinghomes_inage.pdf). Even though these findings are only partially reliable, they clearly show that mortality in nursing homes has been very high and significantly contributed to the total amount of COVID-19 deaths in Italy. However, further studies are needed to capture to impact of such results at a regional level. The text of the manuscript has been updated in the limitation of the study paragraph taking into account this consideration.
- Could the number of deaths be affected by the lack of a laboratory diagnosis of Covid-19? Many deaths in places and periods of increased virus spread, due to lack of swab diagnosis, maybe underestimated, while deaths in terminal patients may have been attributed to Covid-19.
This comment is especially true considering the elderly patients in the nursing homes. However, generally patients affected by COVID-19 are tested when hospitalized or when visited in the emergency room.
- Have the authors evaluated whether the territorial medicine in the different regions has had a different impact on the number of recognized cases in the function of the organization and the availability of nursing and medical staff and adequate diagnostic tools and personal protective equipment?
I think the territorial system was one of the keys of success to limit the hospitalization of patients with COVID and confine them in their home. Care for patients at home or early discharge need not only the availability of nursing and medical staff and adequate diagnostic tools and personal protective equipment robust territorial system, but also the integration of primary, secondary and social care especially at the beginning of the epidemic spread when the majority of the health professionals did not know how to treat this pathology.
- Based on the available data, what hypotheses can the authors formulate which interpret most fully the difference in the number and severity of events between regions with low "incidence" (i.e. Basilicata, Campania, Calabria) and those with the highest (i.e. Lombardy, Piedmont, Trento): relationships between public and private health systems, territorial health organization, geographical location, social structure, population density, other.
At the moment there is a lack of studies that analysed this aspect, in particular at regional level. In the results of another study our team shows that this high variability is not present in Germany and France, while a similar result is reported in Spain. It is interesting also to consider that, at the moment, this variability is lower than in the first period. One aspect that can have influenced the transmission of the virus between the north and the south of Italy is related to the measures adopted by the Italian government in the first phase of COVID‐19 outbreak that contributed significantly to the flattening of the epidemic curve with reduction of new cases and consequent lightening of the care response borne by the health service (Lombardo et al., The Italian national survey on Coronavirus disease 2019 epidemic spread in nursing homes, 2020. Available from: https://onlinelibrary.wiley.com/doi/full/10.1002/gps.5487).

Reviewer 2 Report
Dear Authors please see attached document.
Please respond to each observation using the attached document as a reference.

Author Response
Dear Reviewer,
We thank you for revising our paper submitted to the International Journal of Environmental Research and Public Health. We are grateful for your suggestions and comments.
We have answered the comments raised in a point-by-point format addressing any concerns. Comments are reported in red in the following. The manuscript has considerably improved by these modifications and we thank you for your attention. We have highlighted the changes in the main text using track changes in Word.
Yours Sincerely,
Fabrizio Pecoraro
- Introduction: It is necessary to describe global data from covid-19, (cases and deaths) to date.
We added the following description of global data at the end of the paragraph: Although to date the number of cumulative cases is more similarly distributed among regions, the spread of the virus remains prevalent in the northern regions (more than 60%) and, in particular, in Lombardy (more than 17%).
- Introduction: Join in a single paragraph. Explain more clearly the objective of the study
The text has been updated taking into account the reviewer's suggestion. In particular the objective of the study has been expanded adding the following paragraph: This distinction helps correlating the different phases with the patient management approach adopted in each region aiming to explore whether and to what extent the organization of health delivery has had an impact on the subsequent phases in terms of containment and monitoring of cases. This may also outline lessons learned prompting appropriate changes in the future.
- Define how the three groups are defined (<250, <600, >740).
We adopted the Excel conditional formatting feature that considers three clusters of countries depending on the total number of cases: 1) countries with number of cases lower than 33% of the maximum number of cases (i.e. 884 in Val d'Aosta); 2) countries with number of cases between 33% and 66% of the maximum number of cases; 3) countries with number of cases higher than 66% of the maximum number of cases. We modified the text to make this classification clearer.
- Figure 2, include the number of deaths.
The figure has been updated taking into account the reviewer's suggestion
- It is necessary to refer to the official data (civil registry or Italian Civil Protection Department) of registered deaths with data from previous years. Contrast with data reported by health institutions with national or regional death registries (official government website or Ministry of Health for its periodic report).
We adopted deaths and recoveries as a reference value to capture the spread of the virus along with the number of cases. The aim of this paper is not to compare to what extent the different approaches have influenced the death rate.
- 266-267: Very short paragraph
The text has been updated taking into account the reviewer's suggestion
- Mortality rates are mentioned in the discussion and conclusions, but nowhere in the text is data (table) presented on deaths.
The text has been updated taking into account the reviewer's suggestion. We removed the term mortality and replaced it with death.
- Figure 3: “Vertical lines identify the transition between”. Where are the vertical lines and ranges explained?
The following description is added to explain the meaning of the vertical lines: In particular, the violet line specifies the transition between the initiation and the acceleration phases, the azure one between the acceleration and the deceleration phases and the orange one between the deceleration and the preparation phases.
- Review the wording in general there are paragraphs for short.
The text has been updated taking into account the reviewer's suggestion.
- Figure 3: The ranges of the graphs (regions) (Y) should maintain equal ranges and limits, there is no proper relationship, it confuses the presentation of results.
We agree with the reviewer that equal scales may facilitate the comparison between regions. However, adopting the same scale for all regions makes it difficult to compare the wars, home and ICU trajectories of those regions with a low number of cases. For this reason we prefer to maintain this representation and to add a note that specifies that scales are different.
- Discuss the data (results) with approaches adopted in other countries and continents
Unfortunately, data are heterogeneously collected across countries making it difficult to compare them in particular at a regional level. This is principally evident considering data on hospitalization that are principally reported at a national level. As we consider this as a crucial aspect we added the following sentence at the end of the limitations paragraph: Finally, the different approaches adopted in other countries may be investigated to capture their effect on the virus transmission and death rate. However, data are heterogeneously collected across countries making it difficult to compare the past and the actual situation in particular at considering the hospitalization rate at regional level (Pecoraro et al., The efficiency in the ordinary hospital bed management: A comparative analysis in four European countries before the COVID-19 outbreak. PLoS ONE, accepted with minor revisions).
- The authors refer to China (line 103), but these data are never discussed
This study considers the spread of the virus in Italy during the first phase of the outbreak (from the beginning of February to the end of April). In that moment, the Henan region (as well as other regions in China) already reached the plateau (number of new daily cases around 0). This made it possible to clearly identify that the better regression to be adopted to fit the data is the logistic growth. This comment can be removed as many studies have been published at the present time that confirm this hypothesis. For this reason we rephrased the paragraph as follows: For a finalized epidemic, the curve for the cumulative number of infected cases has been fitted using the logistic growth regression. This model has been adopted in different studies both to represent the COVID-19 outbreak [14-17] and to analyse patterns of epidemic occurred in the past [18].

Round 2
Reviewer 2 Report
Dear Authors,
You have done a great job, congratulations.
Only recommend when I ask to put global data on infections, recoveries and deaths (COVID-19) I mean worldwide, I would think we are about (91M, 51M and 2M) respectively, it would be good to put this in the introduction to date.
A hug.